# Additive Manufacturing of WC-Co Specimens with Internal Channels

**DOI:** 10.3390/ma16113907

**Published:** 2023-05-23

**Authors:** Jindrich Sykora, Michael Sedlmajer, Tim Schubert, Markus Merkel, Lubos Kroft, Ludmila Kucerova, Jan Rehor

**Affiliations:** 1Department of Machining Technology, University of West Bohemia, Univerzitni 8, 301 00 Pilsen, Czech Republic; 2Institute for Virtual Product Development, Aalen University of Applied Sciences, Beethovenstr. 1, 73430 Aalen, Germany; 3Materials Research Institute Aalen, Aalen University of Applied Science, Beethovenstr. 1, 73430 Aalen, Germany; 4Department of Materials and Engineering Metallurgy, University of West Bohemia, Univerzitni 8, 301 00 Pilsen, Czech Republic

**Keywords:** 3D printing, preheating, additive manufacturing, efficient cooling, tungsten carbide, indexable insert

## Abstract

Most material removal in modern manufacturing is currently performed using tools with indexable inserts. Additive manufacturing allows for the creation of new, experimental insert shapes and, more importantly, internal structures, such as channels for coolant. This study deals with developing a process for efficiently manufacturing WC-Co specimens with internal coolant channels with a focus on obtaining a suitable microstructure and surface finish, especially inside the channels. The first part of this study covers the development of process parameters to achieve a microstructure without cracks and with minimal porosity. The next stage focuses solely on improving the surface quality of the parts. Special attention is given to the internal channels, where true surface area and surface quality are evaluated, as these characteristics greatly influence coolant flow. To conclude, WC-Co specimens were successfully manufactured and a microstructure with low porosity and no cracks was achieved and an effective parameter set was found. We have developed a process that produces parts with a surface roughness comparable to those of standard SLS manufacturing of steel parts, while still providing a high-quality internal microstructure. The most suitable parameter set resulted in a profile surface roughness of Ra 4 μm and Rz 31 μm and areal surface roughness of Sa 7 µm and Sz 125 µm.

## 1. Introduction

Ever-increasing demands by industry on the performance of components require the application of advanced materials, such as nickel or titanium alloys. These materials provide superior properties in the form of high tensile strength, creep resistance, corrosion resistance, and in the case of titanium alloys, exceptional tensile strength-to-weight ratio [1,2]. These are in the ISO 513 S group of machined materials and their signature properties greatly reduce their machinability. The combination of high strength at elevated temperatures, low thermal conductivity and the presence of hard particles and severe work hardening creates a combination of high thermal and mechanical load on the cutting tool. Moreover, these materials usually show a tendency to create a built-up edge. For these reasons, regardless of the cutting material used, it is crucial to provide the most efficient cooling of the cutting edge to ensure good tool life [3,4,5].

Additive manufacturing (AM) has recently gained significant attention due to its numerous advantages over traditional manufacturing methods. AM is efficient in producing customised and sophisticated products with complex geometry, reducing material waste, energy consumption, and lead times. AM technologies are also considered environmentally sustainable, producing fewer CO_2_ emissions and enhancing the circular economy. This makes AM a promising solution for addressing sustainability challenges and improving the efficiency and effectiveness of manufacturing processes [6,7].

Efficient coolant delivery is critical for ensuring good tool life when machining ISO S materials. Additive manufacturing allows for the creation of intricately-shaped internal channels that can deliver coolant to the cutting edge. These channels can be easily shaped in a way that minimizes energy loss. Furthermore, the shape of the nozzle can be designed to increase the velocity of the coolant and to cover with coolant the whole used portion of a cutting edge for various insert shapes. Additive manufacturing is often the only option for manufacturing complex internal features, but it is also highly economical for complex features that could be manufactured conventionally. In contrast to machining, the cost of additively manufactured parts does not grow with the complexity of the features [8]. Furthermore, the use of additive manufacturing significantly lowers the overall emissions produced by the whole manufacturing chain [9].

At the moment, the most widely used material for cutting tools is tungsten carbide. It is used not only for the parts of the tools that do the actual cutting, but also for supporting parts, such as shanks of boring bars, top clamps for lathe tools or insert seats, as this material provides higher stiffness and wear resistance than tool steel [10]. Tungsten carbide parts manufactured conventionally using powder metallurgy have quite limited geometrical complexity; however, tungsten carbide can be additively manufactured using various methods. The main differences between them are precision, the form of the raw material, and postprocessing or heat treatment that has to be carried out to the parts. Fused deposition modelling (FDM), also known as fused filament fabrication, requires raw material in the form of a string or pellets where the tungsten carbide powder is embedded in a polymer. This material is then melted in the nozzle and selectively deposited. One of the main shortcomings of this method is the high roughness that is caused by relatively thick layers and uneven deposition, which is usually caused by uneven extrusion. The uneven extrusion can also generate macroporosity in some areas. After debinding and heat treatment, the parts manufactured using this method show mechanical properties similar to conventionally manufactured tungsten carbide [11]. Another option is the binder jet method which uses tungsten carbide in the form of powder. It deposits each layer on the build platform using a roller, and then selectively solidifies it with a binder using a printhead. This method is capable of depositing significantly smaller layers than FDM, and thus creates lower surface roughness. Furthermore, binder jetting generates parts with lower macroporosity than the green body and the precision of the parts is also higher. The heat treatment procedure is similar to FDM [12]. The main advantage of both of these methods is that the material is not thermally influenced during the manufacturing process and the sintering process can be precisely controlled. This means that the mechanical properties of tungsten carbide parts manufactured with these two methods are similar to those manufactured using regular powder metallurgy.

Selective laser sintering (SLS) is a widely used method of metal additive manufacturing [13]. This process offers several key advantages over other methods, including the elimination of the need for complex post-processing techniques, such as debinding and sintering. Additionally, the precision, surface quality, and resolution of the parts produced are superior, and porosity can be minimal [14]. This method can be a good option for the rapid prototyping of tungsten carbide parts; however, the process parameters must be precisely controlled to eliminate defects, such as pores and cracks and provide a desirable microstructure [12]. Controlling laser sintering of a WC-Co (tungsten carbide-cobalt) powder is complex and challenging due to the uneven energy input and significant thermal gradients [15]. As a result of this complexity, research has been directed toward modifying the process parameters in order to only melt the low melting point phase (such as the cobalt binder). However, this approach inherently leads to porosity that is usually addressed by using infiltration of low melt point metals or hot isostatic pressing (HIP). These post-processing techniques add an additional step to the manufacturing process, resulting in increased cost, time, and energy consumption [16,17]. Furthermore, research clearly shows that brittle phases and tungsten carbide grain growth occur even when low volumetric energy density (VED) is used [18]. Therefore, it is better to set up process parameters to achieve a non-porous structure. It has been shown that if VED is high enough to eliminate porosity, cracks occur due to high thermal gradients. This has been solved by increasing the build platform temperature to 650 °C or higher. This reduces the temperature gradient and the amount of energy required to achieve a dense structure. Consequently, nonporous parts without cracks can be manufactured. However, the structure still shows severe grain growth of the WC and formation of brittle phases [19]. WC grain growth inhibitors (VC, Cr) which are used in conventional powder metallurgy have no effect in this manufacturing process [16]. Fries et al. found that grain growth is significantly less severe in border regions where the laser energy input is lower [20].

The mechanical properties of parts manufactured using this method are typically inferior to those produced through conventional powder metallurgy processes [15,19]. However, for cutting tool components, such as insert clamps and seats with complex coolant channels, the properties may still be sufficient and offer increased stiffness and wear resistance compared to additively manufactured tool steel.

It was found that the shape of the particles of the raw powder has a significant influence on the density of the final specimen. Spherical granules contribute to a higher final density; therefore, raw material produced by spray drying is the most favourable. Using WC-20Co powder produced by this process, Chen et al. manufactured specimens with a density greater than 96% [21].

Additive manufacturing of tungsten carbide parts provides new possibilities for more effective cooling of the cutting edge. However, some methods produce parts that require complex heat treatment and postprocessing, which defeats one of the advantages of AM which is rapid prototyping. Therefore, this work focuses on developing a process that will generate parts that have mechanical properties suitable for parts of tools that do not participate in the cutting action and require a minimum of postprocessing, and the creation of high-quality internal channels.

## 2. Materials and Methods

The additive manufacturing experiments were performed using the SLM 280 machine from SLM Solutions Group AG. This particular unit has a single 400 W IPG fibre laser. Its build platform allows for the use of cylindrical platforms of 90 mm in diameter and preheating of the platform up to 650 °C. Since the last time experiments with additive manufacturing of WC-Co were performed on this machine, it has been upgraded with an enhanced gas flow system [19]. The previous nozzle design was exchanged for a sintered wall which provides more efficient inert gas circulation, and thus a cleaner process environment [22]. Argon was used as a process gas and according to the machine sensors, the oxygen concentration was 0.05% during the first 30 min of the build process, after which it dropped to 0.00%. The order of part sintering was set against the flow of the inert gas.

Based on previous experiments, the build material used was Amperit 526.059 powder, which originally meant the use of thermal spraying. This material, produced by Höganäs Germany GmbH is a pre-sintered WC-Co mixture with 83% of WC and 17% of Co. According to the datasheet, the maximum size of the pre-sintered grain (D90) is 28 μm, mean grain size (D50) 18 μm, and minimum grain size (D10) 12 μm [23]. Tomas et al. reported that the conventional sintering of this mixture resulted in a WC grain size (D99) of 5.8 μm [19]. The relative humidity of the raw powder was measured before processing and kept below 10%. The main advantage of using this pre-sintered powder is its good flowability due to the sphericity of the powder particles and the stability of the particles [19]. Another advantage of this powder is the homogenous phase distribution in the transport container, and thus also in the build chamber. Since it has been shown that the lower sections in the transport container and the build chamber contain significantly more particles with a smaller diameter. For this reason, a non-pre-sintered mixture of WC and Co particles might cause uneven distribution of phases at certain heights of the parts [24].

The samples were prepared using standard metallographic practices; however, they were not etched. They were analysed using an Olympus GX51 metallurgical microscope. The surface roughness was measured optically using a Keyence WHX-6000 with VH-Z100R lens and 800× magnification. The hardware filtration was applied according to ISO 4288. Parameters Sa and Sz were measured in accordance with ISO 25178-2. The profile roughness parameters Ra and Sz inside the internal channel were evaluated according to ISO 4287. The samples were sandblasted (using GMA ClassicCut 80 medium at 0.6 MPa) before measuring the surface quality to remove the remains of partially sintered powder material from the surface, which would influence the readings.

The initial samples show significant aperiodicity of the surface, even though the layer-by-layer nature of the process might suggest otherwise. For this reason, optical roughness was measured and areal surface roughness parameters Sa and Sz were evaluated for each sample. The evaluated area was 2 mm × 0.5 mm and the long side of the rectangle was perpendicular to the build platform. A Gaussian filter was used with λs 25 μm and λc 2.5 mm. Each evaluated face had the same orientation on the build platform to eliminate the possible influence of the direction of the inert gas flow. The effect of sample positioning on the platform and potential influence of spatter fallout was not evaluated.

This research is divided into sequential steps that can be followed to achieve efficient manufacturing of WC-Co specimens, regardless of variations in machine tools and raw materials, among other factors. Figure 1 illustrates the flow chart of the process employed.

## 3. Results

The goal of this research was to create a process for additively manufacturing wear-resistant parts with internal channels. The focus was on the surface quality and true cross-section area of the channels, as this greatly influences the energy loss when the cooling medium flows through the channel. Although the previous research carried out in this area by Tomas et al., Kumar et al., and Bricín et al. was a great starting point, a complete material study had to be performed due to the specific characteristics of each machine [17,19,20]. For example, the gas flow system designs vary significantly between the machines and this influences the whole build process.

The starting point for experiments in this paper was the parameter set used in research carried out by Tomas et al. This initial experiment consisted of a scanning speed of 360 mm/s, laser power of 200 W, hatching distance of 0.045 mm, and a layer height of 0.05 mm. The resulting VED was 267 J/mm^3^. The build platform was preheated to 650 °C [19]. However, all of the parts manufactured with this parameter set showed extensive macro surface roughness, waviness and undesirable edge rounding, and most of the parts also developed macroscopic cracks. These results were considered highly undesirable (see Figure 2).

After consideration of all the inputs, there were two main differences from the research carried out by Tomas et al. The first is the batch of raw material used. Even though it was the same product from the same manufacturer, Tomas et al. found that their raw material contained smaller particles (see Table 1). This could be a consequence of using a powder from the bottom of a transport container as demonstrated by Zetkova [24]. It was demonstrated by Meier et al. that powder particle size can play a significant role on part quality when using the SLS method [25]. The second significant difference was the gas flow system. In contrast to the research carried out by Tomas et al., we used an enhanced gas flow system. This delivers the shielding gas through a sintered wall rather than a nozzle, and thus should provide a cleaner atmosphere in the build chamber as a result. It would take further investigation to verify whether any of these adjustments are responsible for this large difference in results. This initial experiment’s main outcome is that a comprehensive parameter study has to be carried out to determine suitable conditions for additively manufacturing parts from this material.

This research was subdivided into two stages. In the first stage, we developed a parameter set to provide the best possible quality of microstructure in the volume of the specimens. Then, the second stage focused on the improvement of the surface quality and geometry of the specimens. The two distinct parameter sets that were explored are visualised in Figure 3.

### 3.1. Stage 1: Parameter Study with Regard to Microstructure

The first thing to determine was a set of parameters that would provide a satisfactory microstructure in the samples and eliminate cracks that occurred in the initial experiment. It is well documented that additive manufacturing of WC-Co parts using the SLM method causes the formation of brittle η-phases in the structure and severe WC grain growth is also usually present [15,18,19]. Therefore, the study was focused on determining parameter sets that result in a structure with a minimum of cracks and porosity. These are the other two most common defects found in SLM 3D printed parts of WC-Co and can be solved by process parameter development.

The first parameter study consisted of 16 samples with 50 mm/s speed variation and 20 W variation. These parameters were based on previous work carried out by Tomas et al. and preparatory experiments. See Table 2 for the complete set of parameters and volumetric energy density. All of the other parameters were kept constant, namely, the layer height was 0.05 mm and the distance between laser tracks was 0.045 mm.

The wide spread of parameters in this study allowed us to narrow down the sets that might be a good fit for the goals of this study. The first stage of the evaluation was purely visual, it is noticeable that the quality of the surface decreases as the scanning speed increases. Surface quality is also negatively influenced by excessive VED, resulting mostly in undesirable rounding of the edges and severe macroporosity. In this first round, upon visual inspection, eight samples that showed poor surface quality associated with high scanning speed were excluded from metallographic evaluation. The excluded samples are marked with an underline in Table 2. The metallography of the remaining samples was evaluated using optical microscopy. All of the evaluated samples showed minimum porosity and no cracks. Both of the main issues associated with SLM additive manufacturing of WC-Co, grain growth, and brittle η-phase formation, can be observed in all of these samples. Undesirable phases within the cobalt binder region with a distinctly different morphology from the WC grains are shown in Figure 4. Based on the research conducted by Campanelli et al. [26], the round-shaped grains likely represent the η-phase. However, it is not the goal of this research to address these issues.

After evaluation, sample 1A-MK1 with 300 mm/s scanning speed and 210 W laser power was considered the most suitable. This sample showed the best microstructure with minimal porosity and the least cracking. Sample 1A-MK1 also showed the fewest surface defects and lowest macro surface roughness. Since the specimen with the best properties is also a boundary sample, a subsequent parameter study was carried out to investigate the influence of even lower scanning speeds and laser powers (see Table 3).

The samples in the second study that had VED lower than 200 J/mm^3^ showed a large degree of macroporosity, which is visible even to the naked eye. Furthermore, metallographic evaluation of the samples indicates that a VED over 250 J/mm^3^ appears to be necessary to achieve minimal porosity. Similarly to the first parameter study, samples not exhibiting the desired microstructure quality were not metallographically evaluated. Several samples with insufficient VED were evaluated to investigate the influence of lower VED on grain growth, but no significant correlation was found. High magnification revealed that although the microstructure is very similar to the results from the previous study, most of the samples show slightly higher microporosity.

The best parameter sets in this study in terms of microstructure were samples 1C-MK2 (140 W; 200 mm/s; 311 J/mm^3^) and 1D-MK2 (170 W; 200 mm/s; 377 J/mm^3^). They showed similar microstructures to the best parameter set from the first study—1A-MK1 (210 W; 300 mm/s; 311 J/mm^3^). The visual comparison between the best samples from both parameter studies is presented in Figure 5. The comparison of surface quality showed very little difference between the samples; therefore, the parameter set of 1A-MK1 was selected for further experiments. This parameter set provides higher productivity than parameters from the MK2 study, and it also provides a microstructure with minimal porosity and cracks.

### 3.2. Stage 2: Improving the Surface Quality of the Parts

The main goal of this research is to create parts with internal coolant channels. Surface quality inside the channel is the main consideration, as this minimizes friction and energy loss when the cooling medium flows through the channel. Furthermore, it can be very challenging to improve surface quality inside the channels in the postprocessing stage. On the other hand, the outer shell of a produced part can be machined away in the postprocessing stage quite easily, which exposes the volume of the part with a high-quality microstructure that provides the desired wear resistance. For these reasons, contour parameters can be developed with a focus only on surface quality.

In the initial parameter studies, it was observed that lower VED and lower scanning speed tend to provide better surface quality, a finding which corresponds with some current research performed in other materials [27,28]. Consequently, the volumetric process parameters were kept constant with the most suitable parameter set of 1A-MK1 (210 W; 300 mm/s; 311 J/mm^3^) and only contour laser parameters were varied. The aim was to maintain the microstructure that was achieved in previous parameter studies, but improve the surface quality. The default contour parameters were two laser tracks with the laser parameters identical to the volume of the part. These original settings resulted in surface roughness of Sa 35 μm and Sz 542 μm. Severe edge rounding and partial sintering of the surrounding powder were observed as a result of excessive VED.

All of the samples in this stage of the study featured circular and teardrop shaped internal channels with varying cross-section areas. All of the channels were printed perpendicular to the build platform and ranged from 0.8 mm in diameter to 1.4 mm (see Figure 6). However, first, only the outside faces of the specimens were evaluated. The outside faces were measured rather than the channels to acclerate the evaluation process and find the most suitable contour parameters. Then, only the channels of the best samples from each parameter study were inspected.

The first parameter study was based solely on the variation in scanning speed and laser power (see Table 4). The distance between the laser tracks was kept at 0.045 mm, with two contour laser tracks. This parameter study clearly shows that a slower scanning speed tends to provide lower surface roughness (see Table 5). However, all of the parts show severe edge rounding and areas with partially sintered powder that could not be removed by sandblasting. These are both apparent results of excessive VED. Partially sintered powder not only negatively influences the surface roughness, but also enlarges the contour of the part, and thus reduces the surface area of the coolant channels. Even when the lowest contour VED was used, partial sintering and edge rounding occurred. Therefore, it is reasonable to assume that the volumetric parameter set greatly influences the surface when only two contour laser tracks are used.

For this reason, in the second study, the scanning speed and laser power were kept constant at 110 W and 210 mm/s, as these parameters resulted in the best surface quality in the previous step. The number of perimeter laser tracks varied together with the distance between the laser tracks (see Table 6) to minimize the influence of volumetric parameters and further lower the VED of the contour. Both the increase in the number of perimeter laser tracks and the distance between the tracks increases surface quality (see Table 7). As predicted, the resulting porosity in the perimeter of the part is significantly higher with lower VED; however, this should not severely influence the overall performance of the parts when the outside surface is finished by grinding in the postprocessing stage. The machining allowance will depend on the parameter set. When using the parameters that resulted in the best surface quality (six border laser tracks spaced 0.1 mm from each other) at least 0.7 mm grinding allowance should be used to obtain an outside surface with the best possible microstructure quality and mechanical properties according to the metallography study (see Figure 7). The resulting porosity at the perimeters of the part was between 1.97% and 4.04%, while porosity in the volume of the specimen remained between 0.06% and 0.1%. Figure 8 displays the comparison between the microstructure in the contour and volume of the specimen.

The cross section of the internal channels was also evaluated and Figure 9 clearly shows the negative influence of the high surface roughness and excessive VED. A single cross section was made through the specimens 2 mm from the build plate. The real cross-section area is significantly reduced due to partial sintering of powder to the walls and severe roughness and waviness of the surface, which can be seen on Figure 10. When looking at the largest circular channel with a programmed cross-section area of 1.54 mm^2^, the real value was reduced by 48% in sample 1A-MK3, while sample 3D-MK4 with improved contour laser parameters showed only a 26% reduction in surface area. It is clear from the data (see Table 8) that the percentage of area reduction increases as the programmed surface area decreases due to the larger influence of surface roughness. This fact has to be taken into account when designing channels to achieve the desired flow rate through the channel.

Finally, surface roughness inside the largest circular channel was measured. A cut through the centre of the channel (see Figure 10) was carried out using wire EDM and profile roughness was evaluated. Sample 1A-MK3 showed Ra 21 μm and Rz 142 μm and sample 3D-MK4 Ra 4 μm and Rz 31 μm. The sample with improved contour laser parameters shows an 80% reduction in Ra and 78% reduction in Rz inside the coolant channel.

## 4. Discussion

It is clear from previous research that additive manufacturing of WC-Co parts can be particularly challenging. The comparisons of FDM and binder jet methods with the SLS method show that the most significant difference is the necessity for further treatment of additively manufactured parts. This can be a drawback or an advantage depending on the point of view. The treatment required for the FDM and binder jet methods is a complex additional step that adds a significant cost. On the other hand, specimens manufactured in this way can in theory provide a higher quality of microstructure, as the final processing can be similar to conventional power metallurgy.

In this study, the SLS method was used, and a framework was presented for developing a process for additively manufacturing WC-Co parts. The first stage of this research was focused on achieving a high-quality microstructure in the volume of the part. The microstructure was mostly comparable with the results achieved by Tomas et al., although it is difficult to draw a precise comparison due to the significant differences between the two studies [20]. The final porosity (0.1%) is lower when compared with the work carried out by Bricín et al. and Chen et al. [18,19,21]. The slight difference in the results when compared with Tomas et al. is probably caused by a combination of two factors—using a higher VED and a different gas flow system in the machine. The differences in the experimental setup when compared to Bricín et al. and Chen et al. were numerous; however, the most significant factors that resulted in the better outcome could be higher VED and different raw material. However, the intrinsic limitations of the SLS process for AM of WC-Co were not overcome. Severe grain growth can be observed in the microstructure as well as the formation of brittle phases. However, Tomas et al. demonstrated that successive heat treatment could reduce the presence of brittle phases [19].

The second stage was focused solely on the improvement of the surface quality of the specimens. To the best of our knowledge, this topic has not been extensively explored in previous research, despite low surface roughness being a key factor for many applications. In this example, internal channels with cross-section areas up to 1.64 mm^2^ are manufactured. High surface roughness would severely impede the performance of the cooling system by increasing energy loss. Volumetric parameters that provide the best microstructure were fixed and contour parameters were varied. Ultimately, surface roughness comparable with additive manufacturing of tool steel parts was achieved [29]. The severe difference between the surface quality of specimens manufactured in Stage 1 and optimised contour parameters can be clearly observed in Figure 9. The quality of the internal channels was greatly improved. However, the drawback is severe porosity in the contour region. The presence of this porosity on the external surfaces of the parts may impede their performance. To eliminate this, it is recommended to conduct postprocessing, such as grinding, that removes the surface layer from the functional areas. By using this approach, a part with high surface quality inside the channels and minimised energy loss when a medium flows through them can be manufactured, while the wear resistance of the parts is not compromised.

Previous research has proposed that elevating the build platform temperature could enhance the properties of fabricated parts [20]. In the current study, the maximum temperature was constrained to 650 °C due to the limitations of the equipment employed. Further investigation of this hypothesis through future studies with even higher build platform temperatures could be conducted.

## 5. Conclusions

Experiments described in this work show the potential to additively manufacture WC-Co parts with internal coolant channels using the SLS method. Satisfactory microstructure, and thus mechanical properties for many applications were achieved (hardness 946 HV1). One of the main challenges while manufacturing specimens with internal channels is the surface quality, which could reduce the efficiency of coolant delivery. In this paper, the manufacturing procedure for significant improvement of surface quality is developed. Surface roughness Sa was reduced by 65%, profile roughness Ra inside the coolant channel was reduced by 80%, and Rz by 78%. However, the downside of this manufacturing process is higher porosity near the surface of the part, but this surface layer can be removed by grinding to maintain good mechanical properties.

The inherent disadvantages of this manufacturing process, such as severe WC grain growth, do not allow for the use of these parts as cutting tools directly. Their properties make them suitable as cutting tip holders, top insert clamps or insert shim seats. All of these parts can be improved by inserting internal channels for coolant. Even though these parts can be also additively manufactured out of tool steel, carbide additive manufacturing would provide higher stiffness and wear resistance. These factors should significantly increase the lifetime of the parts as well as the cutting edge itself.

To conclude, a manufacturing process which can produce WC-Co parts does not require any form of press tooling or conventional sintering. Furthermore, it allows for the creation of complexly shaped internal channels for coolant. Future studies could investigate the use of this process for manufacturing specimens with more complex internal channels. These might be cutting tool components with an increase in coolant delivery efficiency, which will be investigated in the future. This might bring a significant increase in tool life in ISO 513 S and M material groups, and substantially reduce the environmental impact of the machining process. Additionally, it may be beneficial to conduct further studies exploring the mechanical properties of samples manufactured using the suggested procedure.

## Figures and Tables

**Figure 1 materials-16-03907-f001:**
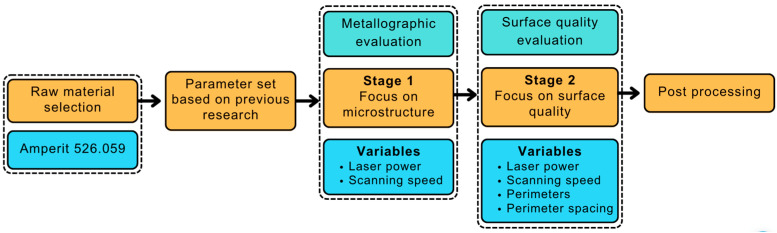
Flow chart describing the entire development process.

**Figure 2 materials-16-03907-f002:**
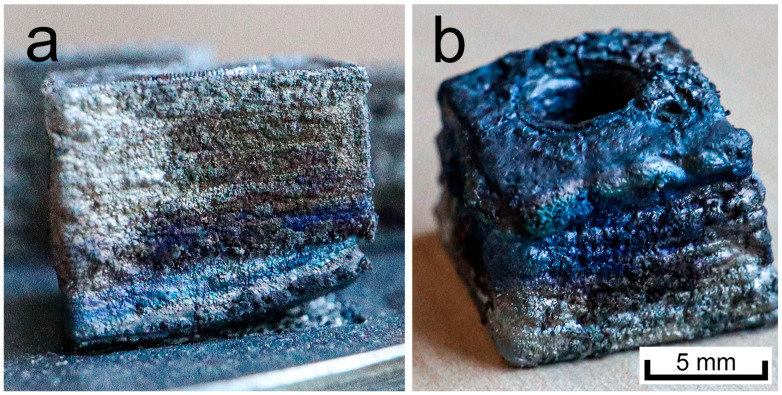
Specimens manufactured using the original parameter set show severe macro roughness, waviness, and cracks. Example (**a**) shows a sample still partially attached to the build platform, partial separation is a result of the build process. Example (**b**) shows the sample removed from the build platform and placed upsidedown.

**Figure 3 materials-16-03907-f003:**
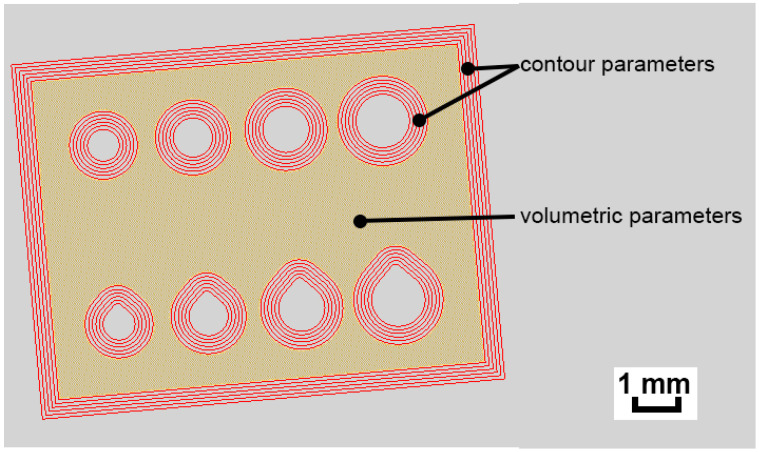
Comparison of contour and volumetric parameters used in the study, the figure illustrates the distinct parameter sets used.

**Figure 4 materials-16-03907-f004:**
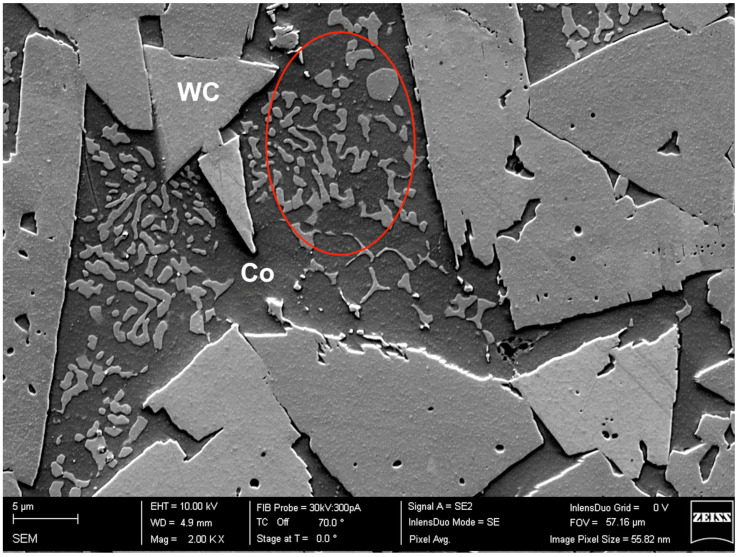
Phases within the cobalt binder region (highlighted by a red circle) with a distinctly different morphology from the WC grains can be observed here.

**Figure 5 materials-16-03907-f005:**
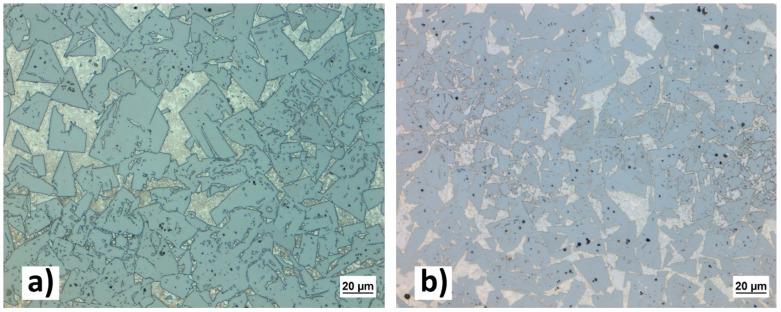
A higher microporosity of sample (**b**) 1C-MK2 can be observed than in sample (**a**) 1A-MK1. Both samples show severe growth in the WC phase.

**Figure 6 materials-16-03907-f006:**
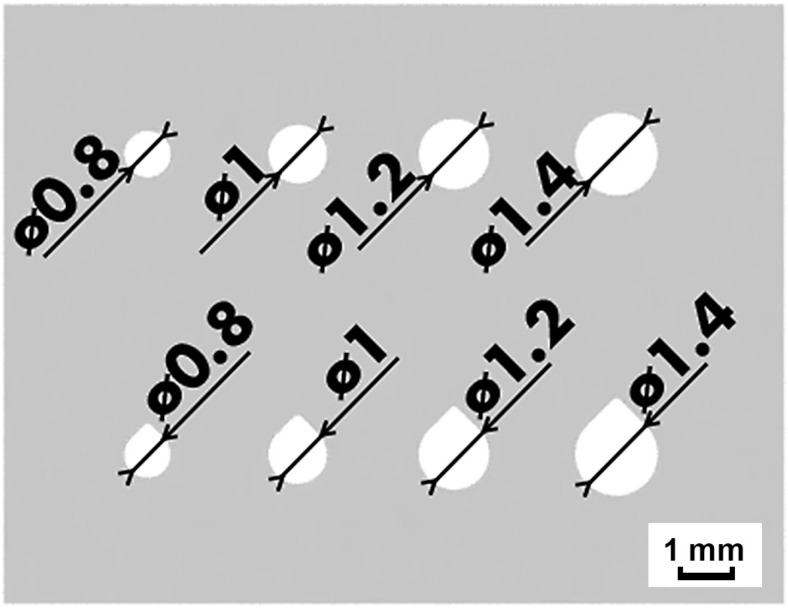
Top view of the test samples used for parameter study in MK3–MK5.

**Figure 7 materials-16-03907-f007:**
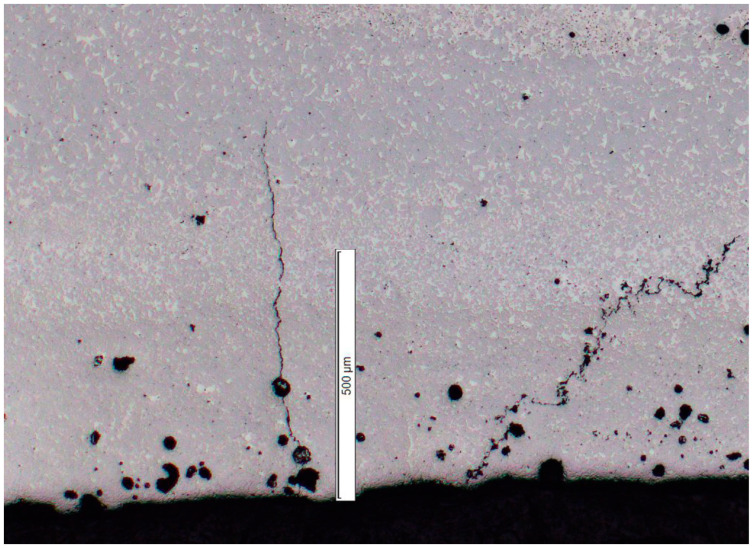
Evaluation of outside perimeter of sample 3D-MK4 clearly shows that at least 0.7 mm should be removed in the postprocessing stage to ensure better mechanical properties of the final part.

**Figure 8 materials-16-03907-f008:**
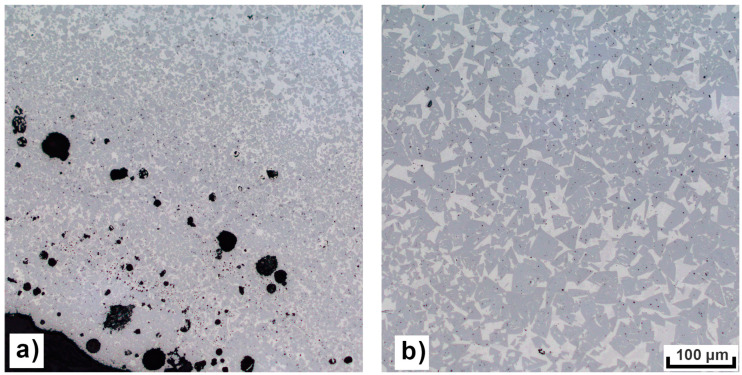
Microstructure comparison between perimeter area around one of the channels (**a**) and volume of the part (**b**). Significant increase in grain size and decrease in porosity can be observed with the volumetric parameters.

**Figure 9 materials-16-03907-f009:**
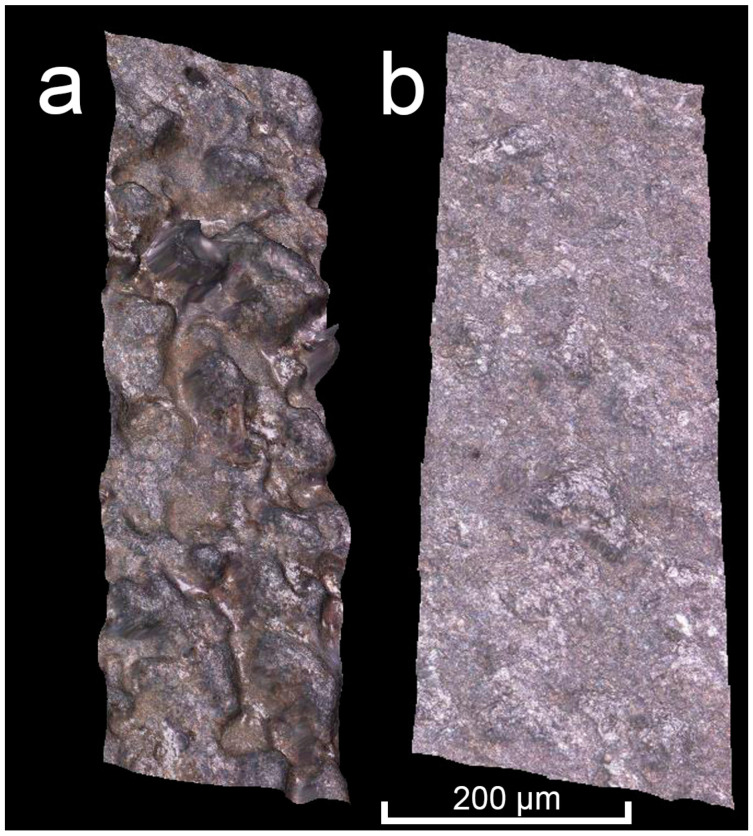
Comparison of surface topography of samples (**a**) 1A-MK3 and (**b**) 3D-MK4.

**Figure 10 materials-16-03907-f010:**
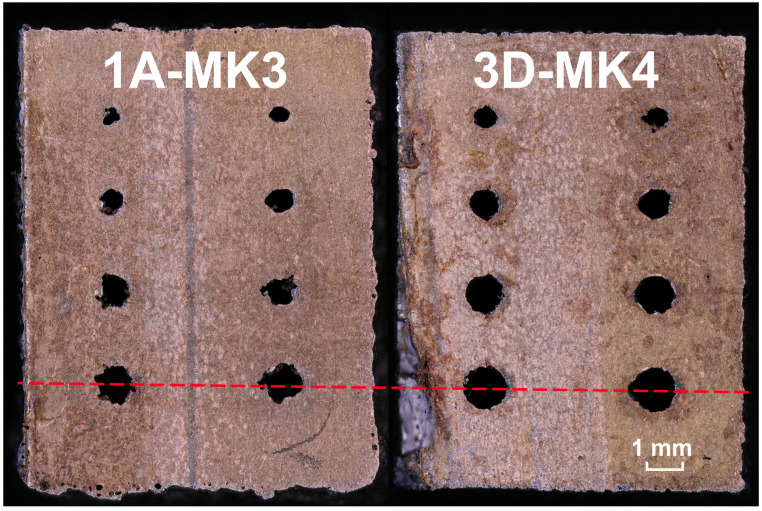
Comparison of internal channel cross section and geometrical accuracy. Red dashed line displays the cut made through the specimens to analyse the channels.

**Table 1 materials-16-03907-t001:** Comparison of the raw material batches used in this study (A) and material analysed and used in work carried out by Tomas et al. (B) [19].

Title 1	A	B
Co [%]	17.7	17.6
C [%]	5	5.1
W [%]	Balance	Balance
Grain size—D90 [μm]	28	-
Grain size—D50 [μm]	18	10
Grain size—D10 [μm]	12	-
Grain size—D99 [μm]	-	25

**Table 2 materials-16-03907-t002:** Variable parameters used in the first study with the resulting VED (J/mm^3^)—Study MK1—samples excluded from further evaluation are marked with an underline.

		P [W]			
Sample ID		A	B	C	D
	V [mm/s]	210	230	250	270
1	**300**	311	341	370	400
2	**350**	267	292	317	343
3	**400**	233	256	278	300
4	**450**	207	227	247	267

**Table 3 materials-16-03907-t003:** Parameters used in the consequent parameter study focused on microstructure—Study MK2—samples excluded from further evaluation are marked with an underline.

		P [W]			
Sample ID		A	B	C	D
	V [mm/s]	80	110	140	170
1	**200**	177	244	311	377
2	**250**	142	195	248	302
3	**300**	118	162	207	251
4	**350**	101	139	177	215

**Table 4 materials-16-03907-t004:** Variation of the contour parameters and resulting VED (J/mm^3^), volumetric parameters were 1A-MK1 (210 W; 300 mm/s; 311 J/mm^3^)—study MK3.

		P [W]			
	V [mm/s]	A	B	C	D
1	**210**	**110**	**120**	**130**	**140**
		233	254	275	296
2	**260**	**130**	**140**	**150**	**160**
		222	239	256	274
3	**310**	**150**	**160**	**170**	**180**
		215	229	244	258
4	**360**	**170**	**180**	**190**	**200**
		210	222	235	247

**Table 5 materials-16-03907-t005:** Surface roughness of the samples—study MK3.

		A	B	C	D
**1**	**Sa [μm]**	20	37	33	49
	**Sz [μm]**	243	413	345	487
**2**	**Sa [μm]**	57	62	33	37
	**Sz [μm]**	521	599	442	420
**3**	**Sa [μm]**	54	46	44	48
	**Sz [μm]**	611	463	502	596
**4**	**Sa [μm]**	59	56	58	41
	**Sz [μm]**	541	714	616	452

**Table 6 materials-16-03907-t006:** Outline parameters used in the study MK4 (volumetric parameter set was 1A-MK1) where X is the number of outline laser tracks and Y is the distance between the laser tracks. Laser power was fixed at 110 W and scanning speed at 210 mm/s.

		A	B	C	D
**1**	VED [J/mm^3^]	233			
	X	3	4	5	6
	Y [mm]	0.045			
**2**	VED [J/mm^3^]	150			
	X	3	4	5	6
	Y [mm]	0.07			
**3**	VED [J/mm^3^]	105			
	X	3	4	5	6
	Y [mm]	0.1			

**Table 7 materials-16-03907-t007:** Surface roughness of the samples—study MK4.

		A	B	C	D
**1**	**Sa [μm]**	29	20	33	12
	**Sz [μm]**	318	296	361	175
**2**	**Sa [μm]**	56	23	17	13
	**Sz [μm]**	472	340	243	245
**3**	**Sa [μm]**	29	19	16	7
	**Sz [μm]**	425	283	212	125

**Table 8 materials-16-03907-t008:** Comparison of designed channel cross section and actual cross section.

	Circle			Droplet		
	Model	1A-MK3	3D-MK4	Model	1A-MK3	3D-MK4
Diameter [mm]	Surface Area [mm^2^]					
1.4	1.54	0.81	1.14	1.64	0.85	1.26
1.2	1.13	0.60	0.86	1.20	0.50	0.93
1	0.785	0.29	0.53	0.84	0.40	0.58
0.8	0.5	0.12	0.3	0.54	0.18	0.26

## Data Availability

Upon request, the corresponding author can provide the original measurement data.

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
