# Peer review of "Additive Manufacturing of WC-Co Specimens with Internal Channels"

_materials, 2023, doi:10.3390/ma16113907_

Round 1
Reviewer 1 Report
The gave a method for efficiently manufacturing WC-Co specimens with internal coolant channels and evaluated the surface and the coolant flow in the channels, Some problem should be noted as bellowing,
(1) The contents about the description of process parameters set in this study are better moved to the section 2-materials and methods, such as table 2, table 3 and table 4.
(2) There is some errors about the format. The stage 2 should be listed as 3.2.
(3) The authors presented two microtopography figure 2. The study of quantitative evaluation about density, strength, surface roughness, component content, the shape and accuracy of the channel, etc. is lack in this section.
(4) There are too much description about the experiment in the section of “result”.
(5) It suggests the author to add some the regularity of laser additive manufacturing parameters on the surface or channel and the characterization to the channel.
The English language is need to be moderated so as to give the prominence to the key points.
Author Response
Thank you for taking the time and going in-depth through the paper. We recognize how much effort it takes to provide such a detailed review and we are grateful for helping us to improve the quality of this work.
(1) The contents about the description of process parameters set in this study are better moved to the section 2-materials and methods, such as table 2, table 3 and table 4.
Originally, we had planned to include the parameter description in Section 2. However, we found that since the parameters and their results are closely related and influence subsequent parameter choices, separating them may complicate the reader's understanding of this progression. This was evident during the initial proofreading. We appreciate your suggestion and have considered it carefully. However, we believe that keeping the sequence of parameters and results as they are would better serve the clarity of the paper.
(2) There is some errors about the format. The stage 2 should be listed as 3.2.
We went through the text, and solved all the issues found.
(3) The authors presented two microtopography figure 2. The study of quantitative evaluation about density, strength, surface roughness, component content, the shape and accuracy of the channel, etc. is lack in this section.
Based on the description provided, we believe that the comment relates to Figure 1, which displays the results of the initial test. We appreciate your input, and in response to your suggestion, we have added a scale to enhance the informative value of the figure. We omitted the quantitative evaluation in this section due to some noticeable issues with the specimen, such as low surface quality, edge rounding, and cracks that were visible to the naked eye.
(4) There are too much description about the experiment in the section of “result”.
We have taken your comment into consideration and made changes to the manuscript. Specifically, we have moved the description of the methods used for surface evaluation to Section 2, which we believe makes for better readability. While we carefully considered moving other passages as well, we have decided to keep the sequence of parameters and results as they are, as we believe this better serves the clarity of the paper in accordance with our reaction to comment (1).
(5) It suggests the author to add some the regularity of laser additive manufacturing parameters on the surface or channel and the characterization to the channel.
We appreciate your feedback and would like to clarify your suggestion. Could you please provide more details or an example of what is meant by this? This would help us better understand your suggestion and make appropriate revisions to the manuscript.
Comments on the Quality of English Language
The English language is need to be moderated so as to give the prominence to the key points.
Thank you for the suggestion, we now had the manuscript checked by a native English speaker and hopefully, the language should now be properly moderated.
Reviewer 2 Report
The authors studied the additive manufacturing of WC-Co specimens with internal channels. The manuscript had an interesting topic and was well-written, however it could only be accepted with the following minor revisions:
· The abstract of the paper remains vague; the conclusion of the study was not found in the text.
· Please check, the text didn't define the WC-Co acronym.
· Abbreviation was found in keywords (e.g.; 3D printing), my suggestion is it could be ‘Three dimensional (3D) printing’.
· It is advised at the beginning of the Introduction Section the authors should define briefly about additive manufacturing (AM) technology and its advantages. Therefore, it is recommended the authors can add the following papers as references:
Design for sustainable additive manufacturing: a review Sustain. Mater. Technol. 35 (2023), e00576
Investigation of ABS–oil palm fiber (Elaeis guineensis) composites filament as feedstock for fused deposition modeling, Rapid Prototyping Journal, 2022, (https://doi.org/10.1108/RPJ-05-2022-0164).
· For the experimental procedure, it is recommended the authors provide a flow chart for the experimental setup.
· Figure 1 didn't locate the scale bar and didn't label each image.
· Refer to line 188, Tomas? Where is the reference/citation?
· Section 3.1 (stage 1) and 3.2 (stage 2) were lacking in writing the discussion of the result obtained and citing the previous studies.
· Figure 6, 7 and 8 did not locate the scale bar.
· The conclusion could be improved by adding the future study, limitations and implications for researchers.
· 40% of the references were outdated (past 5 years). The most recent reference must be updated accordingly throughout the paper.
In this paper, English was fine.
Author Response
Thank you for taking the time and going in-depth through the paper. We recognize how much effort it takes to provide such a detailed review and we are grateful for helping us to improve the quality of this work.
- The abstract of the paper remains vague; the conclusion of the study was not found in the text.
We made changes to the abstract and omitted sections that did not communicate the conclusion clearly, hopefully, it is more definitive now.
- Please check, the text didn't define the WC-Co acronym.
We added the definition to line 109, where the acronym is first mentioned in the body of the text.
- Abbreviation was found in keywords (e.g.; 3D printing), my suggestion is it could be ‘Three dimensional (3D) printing’.
Thank you for the suggestion. We kindly suggest keeping the term as it is referred to in the standard ISO/ASTM 52900:2021(en) Additive manufacturing — General principles — Fundamentals and vocabulary (section 3.3.1. – https://www.iso.org/obp/ui/#iso:std:iso-astm:52900:ed-2:v1:en), which serves as a widely accepted reference in the field of additive manufacturing.
- It is advised at the beginning of the Introduction Section the authors should define briefly about additive manufacturing (AM) technology and its advantages. Therefore, it is recommended the authors can add the following papers as references:
Design for sustainable additive manufacturing: a review Sustain. Mater. Technol. 35 (2023), e00576
Investigation of ABS–oil palm fiber (Elaeis guineensis) composites filament as feedstock for fused deposition modeling, Rapid Prototyping Journal, 2022, (https://doi.org/10.1108/RPJ-05-2022-0164).
We added a section that deals with these topics on lines 58–64.
- For the experimental procedure, it is recommended the authors provide a flow chart for the experimental setup.
Thank you for your suggestion. We would be happy to include a flow chart in the manuscript to clarify the experimental setup. Could you kindly provide us with more information regarding the specific type of flow chart you are referring to? There are several variations of flow charts, and we want to make sure that we include the appropriate one. Could you please provide us with an example or a more detailed explanation of the type of flow chart you suggest?
- Figure 1 didn't locate the scale bar and didn't label each image.
We added a scale bar as well as labels.
- Refer to line 188, Tomas? Where is the reference/citation?
After adding more information, the reference/citation [20] is currently located on line 204.
- Section 3.1 (stage 1) and 3.2 (stage 2) were lacking in writing the discussion of the result obtained and citing the previous studies.
We believe that the results obtained in sections 3.1 and 3.2 are closely related to each other. Therefore, discussing them individually may be either repetitive or unclear. To avoid this, we have included a separate section (4. Discussion) where the results are discussed and compared to previous studies.
- Figure 6, 7 and 8 did not locate the scale bar.
We added scale to all of the figures.
- The conclusion could be improved by adding the future study, limitations and implications for researchers.
We have modified the last paragraph of the conclusion to include this information.
- 40% of the references were outdated (past 5 years). The most recent reference must be updated accordingly throughout the paper.
We updated the references to include more recent research, more than 85% are now references published in the last 5 years.
Reviewer 3 Report
The author's research work is interesting and WC - Co particles dispersion patterns were not discussed in detail. The dispersion mechanism needs to be discussed in brief along with the phase analysis.
Author Response
Thank you for taking the time and going through the paper. We recognize how much effort it takes to provide a review and we are grateful for helping us to improve the quality of this work.
The author's research work is interesting and WC - Co particles dispersion patterns were not discussed in detail. The dispersion mechanism needs to be discussed in brief along with the phase analysis.
We have added information on the morphology and dispersion of the WC-Co particles together with a detailed SEM scan (Figure 4) of one of the samples.
Reviewer 4 Report
The paper needs a thorough review and revision. There are some minor inaccuracies and typos that need to be corrected.
For example, the text mentions "snowed" instead of "showed" in several places.
A reference to two papers on the uneven distribution of powder, one of which is not in English, and the second paper does not mention this topic, is not helpful.
The order of the references in the text could be more systematic, and I would appreciate it if the authors were mentioned with et al. in the text. For example, Tomas, Kumar, and Bricin are referred to in the results. It is not readily evident that these are different papers.
The mentioned VED with respect to the paper by Tomas et al. does not result from the mentioned parameters. How can the massive discoloration of the parts in Fig. 1 be explained?
The particles' size in Tomas et al.'s paper seems to be different; Tomas et al. mention an average diameter D50 of 10 µm. So the size distribution of the particles appears to be different. Tomas et al. mention nitrogen and argon as process gases, but the paper did not specify how these gases were supplied. There seem to be significant differences between the documents.
Based on these experiments, other laser parameters were chosen, and further investigations were carried out, listed in Table 2. According to the text, faulty parts were marked in the table and not investigated further. This marking and which criteria exactly led to the rejection need to be included. If the experiment with VED of 311 J/cm2 has led to the best result, why the range with lower VED was investigated further is unclear. The text mentions that it is clear that the VED must be at least 200 J/cm2 or even that a VED > 250 J/cm2 is needed. Did this only become apparent after the examinations? Please be more specific.
Was the quality in terms of the porosity of the samples only assessed visually? From Fig. 3, one can indeed assume this statement, but is there also a quantitative result for determining the porosity and the grain sizes? This also applies to Fig. 6, for example.
Some figures are not mentioned in the text; figures should support or confirm statements in the text; here, they are often not mentioned. Fig. 7 shows what exactly, on what scale, and what is supported by this figure? If the figure serves no particular purpose, it could be omitted. Fig. 8 should also include a scale. How were the cross-sections measured? Are they single cross-sections, or were several sections made and measured?
To measure the roughness in the channel, was a longitudinal section made through the sample? In the discussion, it is mentioned that a clear growth of the grains and the formation of brittle phases could be observed. Where was the formation of brittle phases investigated in this paper?
Author Response
Thank you for taking the time and going in-depth through the paper. We recognize how much effort it takes to provide such a detailed review and we are grateful for helping us to improve the quality of this work.
For example, the text mentions "snowed" instead of "showed" in several places.
We went thoroughly through the paper and corrected all the typos found.
A reference to two papers on the uneven distribution of powder, one of which is not in English, and the second paper does not mention this topic, is not helpful.
We appreciate your feedback on our reference list. Upon reviewing our manuscript, we found that the topic related to reference [26] was not included in the final version of the text. Therefore, we have removed this reference from our list.
Unfortunately, we were unable to find an English language research paper discussing this specific topic. However, we have changed for the English version of the title and included the exact pages where this topic is discussed in the reference, making it easier for the reader to use an online translation service if necessary.
The order of the references in the text could be more systematic, and I would appreciate it if the authors were mentioned with et al. in the text. For example, Tomas, Kumar, and Bricin are referred to in the results. It is not readily evident that these are different papers.
We added et al. to all places where the authors are mentioned to improve clarity.
The mentioned VED with respect to the paper by Tomas et al. does not result from the mentioned parameters. How can the massive discoloration of the parts in Fig. 1 be explained?
We mentioned the two main differences, namely the gas flow system and a batch of raw powder, in the paper (line 220). Although our first speculation based on the colouration and morphology of the specimens was that excessive VED was the cause, this does not correspond with later results where programmed VED was even higher than in this case. However, since the main goal of our paper is not to reproduce the results achieved by Tomas et al. but to use them as a starting point to develop a process that produces desirable results, we decided not to investigate this issue further and to focus on our main objective.
The particles' size in Tomas et al.'s paper seems to be different; Tomas et al. mention an average diameter D50 of 10 µm. So the size distribution of the particles appears to be different. Tomas et al. mention nitrogen and argon as process gases, but the paper did not specify how these gases were supplied. There seem to be significant differences between the documents.
Thank you for bringing this issue to our attention. We used the raw material data sheet provided by the authors (Tomas et al.) and the manufacturer for our original study. However, we have revised the section (lines 220-226) on the raw powder to include the analysis conducted by Tomas et al. We are somewhat surprised by their findings, as the Amperit 526.059 powder we used in our study is the finest variant supplied by the manufacturer (Tomas et al. also report using Amperit 526). One possible explanation for this difference could be that Tomas et al. used powder from the bottom of the container for their analysis and part production. We have included this information in the revision. Although there are many differences between our research and theirs, our aim was not to replicate their results, but to build upon their findings as a starting point.
Based on these experiments, other laser parameters were chosen, and further investigations were carried out, listed in Table 2. According to the text, faulty parts were marked in the table and not investigated further. This marking and which criteria exactly led to the rejection need to be included.
We added the underline to the rejected samples in Table 2 and updated the explanation for the rejection criteria on lines 270-273.
If the experiment with VED of 311 J/cm2 has led to the best result, why the range with lower VED was investigated further is unclear.
The full parameter study was done to also evaluate the influence of these lower parameters on the surface quality and grain growth. The results were however not included to keep the paper concise and to the point.
The text mentions that it is clear that the VED must be at least 200 J/cm2 or even that a VED > 250 J/cm2 is needed. Did this only become apparent after the examinations? Please be more specific.
We added an updated explanation for this conclusion on lines 295-298.
Was the quality in terms of the porosity of the samples only assessed visually? From Fig. 3, one can indeed assume this statement, but is there also a quantitative result for determining the porosity and the grain sizes? This also applies to Fig. 6, for example.
The quality of the porosity of the samples was assessed visually in the first stage. In the second stage, it was assessed using the proprietary Olympus GX51 software.
The WC grain size was assessed visually using the image scale and compared to results gained from conventional sintering of the powder. Information about the WC grain size from conventional sintering was added to the Materials and Methods section.
Some figures are not mentioned in the text; figures should support or confirm statements in the text; here, they are often not mentioned. Fig. 7 shows what exactly, on what scale, and what is supported by this figure? If the figure serves no particular purpose, it could be omitted. Fig. 8 should also include a scale.
We made sure to mention the figures in the text and explain their purpose. A scale was also added to all of the figures.
How were the cross-sections measured? Are they single cross-sections, or were several sections made and measured?
The cross sections were measured using the Keyence proprietary software and a single cross section was made. We have added this information to the text (lines 446-447).
To measure the roughness in the channel, was a longitudinal section made through the sample?
We have included a red dashed line in Figure 8 to indicate the section cut through the specimens for channel analysis.
In the discussion, it is mentioned that a clear growth of the grains and the formation of brittle phases could be observed. Where was the formation of brittle phases investigated in this paper?
We have added a section (lines 277-280) that shortly discusses this topic as well as an additional analysis that supports the statements (Figure 4).
Round 2
Reviewer 1 Report
The gave a method for efficiently manufacturing WC-Co specimens with internal coolant channels and evaluated the surface and the coolant flow in the channels, Some problem should be noted as bellowing,
(1) There is some errors about the format. 2 in CO2 should be subscript.
(2) There is some references that do not be shown normally, such as line 241, page 6. line 269, page 7.
(3) The conclusion can be improved.
there is no much language issues.
Author Response
(1) There is some errors about the format. 2 in CO2 should be subscript.
We reviewed the document and corrected all the errors that were identified.
(2) There is some references that do not be shown normally, such as line 241, page 6. line 269, page 7.
Thank you for bringing it to our attention. We have revised the referencing, and it is now free of errors.
(3) The conclusion can be improved.
We have made some subtle changes that are expected to enhance the informative value of the conclusion.
Reviewer 2 Report
The authors answered all questions. For a flow chart, it could be a process flow chart or an illustration/diagram of the research methodology. You may refer to a sample below.
The quality of the English language was acceptable.
Author Response
For a flow chart, it could be a process flow chart or an illustration/diagram of the research methodology. You may refer to a sample below.
Thank you for the example, we have included a process flow chart in section 2. Materials and Methods.
Reviewer 4 Report
The paper "Additive manufacturing of WC-Co specimens with internal channels" has been modified significantly. Most of my remarks have been implemented or clarified.
Unfortunately, the pdf document I got for review contains a lot of "Error! Reference source not found", which must be corrected before publication.
Author Response
Unfortunately, the pdf document I got for review contains a lot of "Error! Reference source not found", which must be corrected before publication.
Thank you for bringing it to our attention. We have revised the referencing, and it should now be free of errors.